# Phage S144, a New Polyvalent Phage Infecting *Salmonella* spp. and *Cronobacter sakazakii*

**DOI:** 10.3390/ijms21155196

**Published:** 2020-07-22

**Authors:** Michela Gambino, Anders Nørgaard Sørensen, Stephen Ahern, Georgios Smyrlis, Yilmaz Emre Gencay, Hanne Hendrix, Horst Neve, Jean-Paul Noben, Rob Lavigne, Lone Brøndsted

**Affiliations:** 1Department of Veterinary and Animal Sciences, University of Copenhagen, 1870 Frederiksberg C, Denmark; mgambino@sund.ku.dk (M.G.); anders.norgaard@sund.ku.dk (A.N.S.); s.j.ahern86@gmail.com (S.A.); gsmirlis@gmail.com (G.S.); e.gencay@gmail.com (Y.E.G.); 2Laboratory of Gene Technology, KU Leuven, 3001 Leuven, Belgium; hanne.hendrix@kuleuven.be (H.H.); rob.lavigne@kuleuven.be (R.L.); 3Department of Microbiology and Biotechnology, Max Rubner-Institut, Federal Research Institute of Nutrition and Food, 24103 Kiel, Germany; Horst.Neve@mri.bund.de; 4Biomedical Research Institute and Transnational University Limburg, Hasselt University, BE3590 Diepenbeek, Belgium; jeanpaul.noben@uhasselt.be

**Keywords:** phage, polyvalent, *Enterobacteriaceae*

## Abstract

Phages are generally considered species- or even strain-specific, yet polyvalent phages are able to infect bacteria from different genera. Here, we characterize the novel polyvalent phage S144, a member of the *Loughboroughvirus* genus. By screening 211 *Enterobacteriaceae* strains, we found that phage S144 forms plaques on specific serovars of *Salmonella*
*enterica* subsp. *enterica* and on *Cronobacter sakazakii*. Analysis of phage resistant mutants suggests that the O-antigen of lipopolysaccharide is the phage receptor in both bacterial genera. The S144 genome consists of 53,628 bp and encodes 80 open reading frames (ORFs), but no tRNA genes. In total, 32 ORFs coding for structural proteins were confirmed by ESI-MS/MS analysis, whereas 45 gene products were functionally annotated within DNA metabolism, packaging, nucleotide biosynthesis and phage morphogenesis. Transmission electron microscopy showed that phage S144 is a myovirus, with a prolate head and short tail fibers. The putative S144 tail fiber structure is, overall, similar to the tail fiber of phage Mu and the C-terminus shows amino acid similarity to tail fibers of otherwise unrelated phages infecting *Cronobacter*. Since all phages in the *Loughboroughvirus* genus encode tail fibers similar to S144, we suggest that phages in this genus infect *Cronobacter sakazakii* and are polyvalent.

## 1. Introduction

Bacteriophages (phages) are viruses that rely on their bacterial host for propagation. To produce progeny, phages require perfect compatibility with their host at each step in the infection cycle [1]. The initial binding of the phage to its bacterial host surface is reversible, allowing the phage to proceed into the infection cycle or to detach to bind another host. The interaction only becomes irreversible by complementary and specific binding of the phage receptor binding protein (RBP), at the tail fibers or spikes, to the bacterial receptor. For example, the long tail fibers of phage T4 are known to bind reversibly to a primary receptor on the cell surface, but it is only when three or more tail fibers are bound that the short tail fibers may extend to bind irreversibly to the secondary receptor. This in turn allows DNA injection by sheath contraction and insertion of the inner tail tube into the host cell [2,3]. Once the phage DNA is injected into the host, compatibility in the subsequent processes is required for the phage to exploit the host machinery. For example, a similar codon usage and promoter elements recognized by the host RNA polymerase are regarded as a compatibility trait between phages and hosts [4,5]. Furthermore, phage proteins must be adequately synthesized to form mature and functional phage particles by using the host cell machinery. Overall, the strict compatibility between phage and host required for phages to complete their infection cycle imposes severe restrictions on the host range of phages.

Because of these limitations, the host range of a phage may be very narrow, possibly limited to few bacterial strains. For example, phages of the *Jerseyvirus* genus only infect one or two selected isolates of 21 *Salmonella* Derby tested, demonstrating a very narrow host range [6]. On the other hand, phages with broader host range might be very attractive from an applied point of view, since they could target more pathogenic strains at once [7,8]. Broad host range phages are often isolated from environments with large bacterial diversity, such as wastewater or free-range dairy farms. This diversity promotes genetic exchanges, thus increasing the probability for phages to recombine and expand the set of targeted hosts [6,9]. Even more interesting from an applied and molecular point of view are the so-called polyvalent phages, i.e., phages able to infect bacteria from different genera. Yet, our understanding of their molecular and evolutionary complexity is still limited to few examples [10,11,12,13]. A phage may infect hosts from diverse genera simply by recognizing a common receptor conserved among these bacterial species. For example, phage PRD1 infects a broad range of Gram-negative bacteria, such as *Enterobacteriaceae* genera, but also *Pseudomonas aeruginosa*, that all carry the conjugative IncP plasmid. The plasmid encodes a type IV trans-envelope DNA translocation complex, which is recognized as a receptor by PRD1 [14,15], thus expanding the host range of the phage to any strain able to receive the plasmid. Other phages are able to recognize diverse receptors on bacteria of different genera because they are equipped with multiple RBPs. These phages often have a complex tail structure containing several different tail fibers and/or spikes, each recognizing a different receptor on the host bacterium. For example, in phage Phi92 the baseplate contains radiating fibers and tail spikes resemble an open Swiss army knife. Each phi92 particle carries five different tail spike and tail fiber proteins, each recognizing different receptors, allowing the phage to infect *E. coli* strains with or without capsule, as well as many *Salmonella* serovars [11,16]. Similarly, the recently described function and substrate specificity of three tail spikes of phage CBA120 revealed that each of them is specific for a specific *E. coli* serotype and that the fourth tail spike might be responsible for the infection of *Salmonella* Minnesota [13].

The limiting obstacle for identifying new polyvalent phages is the requirement to perform extensive host range assays, involving a considerable number of strains from each genus. Recent phage host prediction tools might help to narrow down the genera to screen (for example, HostPhinder, [17]), but it is important to keep in mind that such prediction tools also rely on the availability of biological data and the host genome sequence as a reference [18]. An additional difficulty is the extreme diversity within each bacterial species. For example, *Salmonella* spp. accounts for more than 2500 serovars [19], displaying an incredible surface diversity and a consequent diversity in phage sensitivity that it is hard to cover with bioinformatic tools because of a conspicuous number of additional variables.

We previously established a *Salmonella* phage collection by isolating phages from animal and environmental sources using a variety of isolation hosts. During characterization of the phages, we observed that phage S144, isolated from wastewater, was the only phage isolated using *Salmonella enterica* serovar Infantis as isolation host [6]. Moreover, phage S144 was not grouped with other phages within our non-metric dimensional scaling (NMDS) analysis and had a narrow host range on the 72 diverse *Salmonella* strains tested, only infecting nine strains. Here, we screened a large collection of 211 *Enterobacteriaceae* strains and found that phage S144 is a polyvalent phage infecting strains of some *Salmonella* serovars, as well as of *Cronobacter sakazakii* and *Enterobacter* species. Despite the fact that these bacteria all belong to the *Enterobacteriaceae*, they are distinct genera [20,21]. By genome and proteome analysis, we characterize this polyvalent phage and discuss the molecular reasons behind its cross-genera infectivity and reveal its taxonomic position by a comprehensive comparative genomics analysis.

## 2. Results

### 2.1. Phage S144 Is a Polyvalent Virus, Infecting Diverse Strains of the Enterobacteriaceae

Phage S144 was previously isolated from wastewater using *Salmonella enterica* serovar Infantis as isolation host, forming very small and turbid plaques. The phage had a narrow and unusual host range compared to our extended collection of *Salmonella* phages, only infecting nine of the 72 *Salmonella* strains tested [6]. To explore the possibility that phage S144 could infect other genera and thus be polyvalent, its infectivity was tested against 211 *Enterobacteriaceae* strains: 92 *Salmonella* strains belonging to 34 different serovars, the ECOR collection consisting of 72 *E. coli*, 5 other *E. coli* strains, 4 of which are O157:H7, 15 *Cronobacter sakazakii* strains, 7 *Yersinia sp.* strains, 4 *Enterobacter sp.* strains, 5 *Acinetobacter sp.* strains, 3 *Klebsiella sp.* strains, 2 *Proteus sp.* strains, 1 strain from each genera of *Serratia*, *Erwinia*, *Shigella* and *Providencia* (Table A1). Of the 211 strains tested, phage S144 forms plaques on 25 strains (Table 1). The sensitive strains include 16 genera of *Salmonella*, but when more strains of the same *Salmonella* serovar were tested, as Derby, Enteriditis and Infantis, not all could be infected by S144 (Table A2). On top of the 16 serovars of *Salmonella*, phage S144 infects *Enterobacter cloacae* and 4 out of the 15 *C. sakazakii* tested strains (Table 1). Interestingly, completely different plaque morphologies were observed for different hosts. While S144 plaques are small and turbid and very hard to see on *S.* Infantis, the plaques are more visible on *C. sakazakii* and even more clear and large on *S.* Muenster (Figure A1). In addition, we observed that a phage stock propagated on *S.* Infantis infects *Cronobacter* strains and *S.* Muenster with a comparable or even higher titer than *S.* Infantis, suggesting that *Cronobacter* and *S.* Muenster might be more suitable hosts for phage S144 than *S.* Infantis.

### 2.2. S144 Recognizes the O-Antigen as Receptor Both in Salmonella and Cronobacter

To identify the receptor of phage S144 in *S.* Muenster and *C. sakazakii*, colonies of both hosts resistant to phage S144 were isolated after exposure to the phage (see Material and Methods). To confirm that the isolated colonies were receptor mutants, we performed an adsorption assay (Figure A2). While most of the phages adsorbed to the wild types within 10 min (99% on *S.* Muenster and 87% on *C. sakazakii*), no significant reduction in titers of free phages was observed in the mutants, thus confirming that they were receptor mutants. Sequencing analysis of the genome of the resistant *S.* Muenster mutant identified various genes with no reads coverage (gaps) compared to the wild type strain. These were coding for prophage proteins, fimbriae and involved in O-antigen production (Table A3). In particular, the large gap corresponding to the rhamnosyl (*wbaN*) and the mannosyl transferase (*wbaO*), the O-antigen polymerase (*wzy*) and flippase (*wzx*), and the dTDP-4-dehydrorhamnose 3,5-epimerase (*rmlC*) suggest that the mutant lacks the O-antigen of the lipopolysaccharide. Conversely, the missing coverage of fimbriae genes (*yeh*, *yad* and K88) might indicate that phage S144 also depends on fimbriae during binding. Sequencing analysis of the genome of a *C. sakazakii* resistant mutant revealed deletions in the operon for the O-antigen biosynthesis, specifically for the genes coding for four glycosyl transferases, the O-antigen polymerase, the flippase and the aminotransferase (gene products from 744 to 750; Table A4). In addition to the O-antigen biosynthesis pathway, the resistant *C. sakazakii* mutant lacked parts of the operon for capsule biosynthesis (gene products from 1735 to 1738; Table A4), indicating that the capsule formation might be impaired and may also play a role in the binding of the phage. According to the proposed classification, our wild type *Cronobacter* strain belongs to serogroup O2 [22], thus exposing an O-antigen with L-rhamnose and D-galactose. Interestingly, these sugar residuals are also present in the O-antigen of *S.* Muenster (*Salmonella*’s serogroup O3, [23]). Even though further experiments are necessary to demonstrate that these sugars allow the binding to S144 tail fibers, our data suggest that the O-antigen might be needed for phage S144 to infect both bacterial hosts.

### 2.3. Morphology of Phage S144

Electron microscopy observations of phage S144 showed a typical myovirus morphotype, with a prolate head 101.4 ± 4.8 nm long and 43.8 ± 1.2 nm wide (Figure 1). The tail (96.9 ± 2.5 nm long and 20.4 ± 0.7 nm wide, see Figure 1a–c) is also shown with a contracted sheath, and either with intact capsid (Figure 1d) or with empty capsid (Figure 1e), respectively. Phage S144 has short and notably rigid tail fibers, occasionally detected in extended configuration (Figure 1b,c, 25.4 ± 2.3 nm long). These fibers were usually flipped up in retracted position (attached to the tail surface) (Figure 1a).

### 2.4. Functional Modules of Phage S144

To further characterize this polyvalent phage, we sequenced the genome of phage S144 and found that it consists of 53,628 bp of double-stranded DNA with 45.8% GC content and no tRNA synthesis genes detected. The S144 genome is predicted to encode 80 open reading frames (ORFs) with ATG as initiation codon for 73 of the ORFs, followed by GTG for five ORFs and TTG for a single ORF. Many ORFs (39 ORFs, corresponding to the 49%) are predicted as hypothetical proteins, with no similarity to characterized proteins (Table 2 and Appendix A). Interestingly, ORF71 to ORF80, forming a cluster of genes transcribed in the same direction (cluster E in Figure 2), have a lower GC content (38.6 ± 4.6) compared to the rest of the genome (46.1 ± 2.8). This could be due to the typical GC skew pattern (Figure 2) or might be indicative of a horizontal gene transfer event for cluster E, although no function can be assigned to all of these small genes. From the predicted function of the remaining 41 genes, the direction of the transcription and the presence of promoters allowed us to group the genome into four additional clusters. Cluster A (ORF01 to ORF06) codes for enzymes involved in DNA metabolism and nucleotide biosynthesis, cluster B (ORF07 to ORF37) is responsible for the packaging, the phage morphogenesis and the lytic cassette, the cluster C (ORF38 to ORF67) constitutes the main operon for the DNA replication, transcription and encodes also other genes involved in nucleotide biosynthesis and DNA modification, and cluster D (ORF68 to ORF70) is probably involved in the DNA metabolism as well. (Figure 2, Table 2 and Appendix A).

In cluster C (ORF38 to ORF67), genes involved in the replication of the phage genome including the DNA polymerase (alpha and beta subunit, ORF41 and ORF42), an ATP-dependent helicase (ORF51) and an exonuclease (ORF52) were found. Other genes involved in the DNA metabolism are a resolvase (ORF70, cluster D) and the Ku protein (ORF01, cluster A). The resolvase (ORF70) may be involved in the cleavage of branched DNA molecules during the replication or DNA repair processes, but also in the join-cut-copy pathway just before packaging [24,25,26]. ORF01 is predicted to function as a Ku protein, similar to the Gam protein of phage Mu: early transcribed, it binds to linear Mu DNA, thus preventing its degradation by bacterial exonucleases at the very beginning of the phage infection [27].

Since no RNA polymerase was identified within S144 genome, we suggest that S144 utilizes the host RNA polymerase for transcription. A total of 18 promoters compatible with *Salmonella* RNA polymerase were predicted, 7 of which also match promoters from *Cronobacter* spp. (Appendix A). This suggests that the two bacterial genera have conserved promoter motifs that allow phage S144 to exploit both hosts transcriptional machinery. Promoters have been predicted within all clusters. Several promoters precede the lysis cassette (see below ORFs 34–37) and the clusters A and B, in the same directions as the ORFs (Figure 2, Appendix A), indicating that these clusters form also transcriptional operons. While the early promoters may be solely recognized by the host transcription machinery, later expressed genes may be controlled by ORF38 (cluster C) encoding a DksA C4-type domain-containing protein, known as an activator of transcription initiation [28,29]. The translation also relies on the host tRNAs pool, since no tRNA synthesis genes have been identified in S144 genome. We observed that S144 codon usage correlates with the tRNA pool in *S.* Muenster and *C. sakazakii*, with the most required amino acid (leucine, serine, arginine and valine) corresponding to abundant tRNA in the hosts (Figure A3).

Five of the annotated genes, localized in clusters A and C, are involved in the nucleotide biosynthesis (Figure 2 and Appendix A). The deoxycytidylate deaminase (ORF53; KEGG K01493) is a key enzyme in the pyrimidine metabolism. It provides the substrate for the deamination of deoxycytidine monophosphate (dCMP) to deoxyuridine triphosphate (dUMP) [30]. In the same pathway, dUMP is the nucleotide substrate for thymidylate synthase, function predicted for the ORF53, which leads to the production of deoxythymidine monophosphate (dTMP). dTMP is then converted, in an ATP-dependent way, to deoxythymidine diphosphate (dTDP) by the thymidylate kinase [31,32], an enzyme coded by ORF04. Other genes that may support this pathway are a P-loop containing nucleoside triphosphatehydrolase (ORF55), which commonly hydrolyses the nucleoside triphosphate to produce energy [33], and the dihydrofolate reductase (ORF03), which initiates the pathway for the production of 5,10-methylenetetrahydrofolate, a cofactor of thymidylate synthase [34]. Based on these findings, it is reasonable to suggest that S144 produces its own pool of thymidine triphosphate (dTTP) and it is likely that some of the hypothetical genes may encode some of the missing enzymes of this pathway.

The B cluster is composed of 31 genes, from ORF07 to ORF37, involved in the DNA packaging and the morphogenesis of capsid, baseplate and tail (Figure 2, Table 2, Appendix A). Most of them have been identified as structural proteins by electrospray ionization tandem mass spectrometry (ESI-MS/MS) with a sequence coverage ranging from 4% to 80% (Table 2, Appendix A, Figure A5) and are indicated in Figure 2. These included proteins with predicted functions in the structure of the capsid (gp09–gp11), neck (gp12, gp14–gp15), tail (gp16–gp18, gp20–21, gp23), baseplate (gp26–gp30) and tail fibers (gp31), as well as six proteins of unassigned function. The remaining identified proteins have functions in nucleotide biosynthesis (gp03 and gp43) and cell lysis (gp35–gp37). ORF08 is predicted to function as the large subunit of the terminase. No similarities have been revealed for ORF07, but its position and size suggest that it is the small subunit of the terminase.

Since genes from ORF07 to ORF33 code for structural proteins, we further investigated them using homology detection and structural prediction and included all details and scores along with other predicted proteins in Appendix A. Based on these findings, we can propose three modules for phage morphogenesis, namely for capsid, baseplate and tail. ORF09 to ORF15 are likely dedicated to capsid morphogenesis and include the portal (ORF09), the scaffold (ORF10) and the major capsid proteins (ORF11) and four proteins forming the head-to-tail connector complex (ORF12 to ORF15). In the subsequent region, we identified four tail proteins (ORF16 to ORF41), the sheath (ORF42), the tube (ORF21), a chaperone (ORF22) and the tape measure (ORF23). Of the four tail proteins, ORF17 has been annotated as a tail fiber protein in the *Salmonella* phages SE4 and SE13 [35]. Although HHPRED predicts that ORF17 may have hydrolase activity as seen in some tail spikes [36], the PHYRE2 structure prediction did not resemble neither a tail spike, nor a tail fiber (data not shown) and is therefore annotated simply as a tail protein. ORF24 and ORF25 have no similarities with any described genes or proteins, but we suggest they may code for other structural proteins, since they have been identified by ESI-MS/MS and are located in between the genes encoding tail and baseplate proteins. The baseplate is composed of proteins encoded by five genes: four baseplate (ORF26, and ORF28 to ORF30) and a puncturing protein (ORF27; 100% confidence with crystal structure of the bacteriophage phi92 membrane-piercing protein2 gp138 according to PHYRE2).

The genes for the tail fiber morphogenesis include the tail fiber protein (ORF31) and two chaperones (ORF32 and ORF33), which are predicted to be needed for tail fiber assembly, as both proteins showed high homology to GpU, the chaperone of the tail fiber protein of Mu G+ (HHPRED probability 100%, e-value: 1.3^−43^). The structural prediction of ORF31 using PHYRE2 (99.8% confidence), SWISS-MODEL (QMEAN: −3.59) and HHPRED (98.13%, e-value: 2.1e^−8^) indicated homology to the tail fiber from bacteriophage Mu G+ (gpS), more specific the distal C-terminus (amino acids 351–449) responsible for receptor binding [37]. As evident from the superimposed alignment in Figure 3, the PHYRE2 structure prediction of ORF31 aligned exquisitely with the structure of gpS, even though they only share 21.55% identity at the amino acid level. In addition to similarity with other *Salmonella* phage tail fibers, we observed similarity at the nucleotide level of ORF32 with the genes putatively coding for the tail fibers of two *Cronobacter* phages, GAP31 (YP_006987062.1) and GAP32 (YP_006987359.1). This observation was confirmed at the amino acid level, specifically at the distal C-terminus of the tail fiber of S114 (amino acids 351–449) with GAP31 (BLASTp: 71% query cover with 53.62% identity, e-value 5e-15) and GAP32 (BLASTp: 72% query cover with 48.57% identity, e-value 5e-11) (Figure 3A), suggesting a key role of the tail fiber in the recognition and the infection of *C. sakazakii*.

The last four genes of the cluster B code for the lysis cassette needed, for the disruption of the outer membrane, to release the new formed virions [38]. We could find few similarities to the well described phage genes and proteins, yet the position of these genes and the detected domains (Table 2 and Appendix A) indicate that S144 encodes for a holin (ORF34), an endolysin (ORF35) and the inner and outer membrane spanins (ORF36 and ORF37). As in other Type II holins, ORF34 has two transmembrane domains, while ORF35 is recognizable as an endolysin by an O-glycoside hydrolase domain, showing protein to lysins of other phages infecting *Enterobacteriaceae* strains (Table 2 and Appendix A). Since ORF36 has a transmembrane domain at the N-terminal, we classified it as a putative integral inner membrane protein (i-spanin) and we found the outer membrane lipoprotein (o-spanin) at the +1 reading frame (Table 2 and Appendix A), as previously suggested [39].

### 2.5. The Genomic DNA of Phage S144 is Modified

Phage DNA is often modified in order to protect against bacterial restriction and modification systems [40]. To explore this possibility, we extracted the DNA of phage S144 propagated on three different strains (*S.* Infantis S15, *S.* Muenster S394 and *Cronobacter sakazakii* CS1) and digested them with nine different restriction enzymes, each selective for different restriction sites. We found that, similar to the control, namely the methylated DNA of phage Lambda, only the AT-specific restriction enzymes SspI and PacI were able to cut phage S144 genomic (Table A5), thus indicating that phage S144 DNA may be modified on the cytosine. Analyzing the genome of S144, we could not find homology to genes previously described to modify phage/bacterial DNA. Only ORF63 contains a bromo-adjacent homology (BAH) domain that has been identified in eukaryotic DNA (cytosine-5) methyltransferases [41,42]. Conversely, the BAH domain is common in DNA and chromatin-associated proteins. As such, not enough evidence is available to suggest that ORF63 is a methyltransferase. Further research will be necessary to understand the modification of S144 DNA and to identify the genes responsible for it.

### 2.6. Phylogenetic Analysis Shows That S144 Is a Member of the Loughboroughvirus Genus

To gain insight into S144 taxonomy, the S144 genome sequence was submitted to BLASTn. The search (June 2020) revealed high similarity to two *Salmonella* phages deposited in GenBank, the only members of the recent *Loughboroughvirus* genus: SE4 (97% query coverage, 87.42% identity) and ZCSE2 (98% query coverage, 99.08% identity). The next closest relatives are members of the likewise recently accepted *Rosemountvirus* genus, formed by phages infected *Salmonella* too (coverage 70–83%; identity 77–78%). Similar to phage S144, these *Salmonella* phages have a GC content of 45–46%, similar genome size (between 53,494 and 53,965 bp) and no tRNA synthesis genes. The percentage of coding genome (from 92% to 93.4%) and the number of coding sequences is slightly variable (from 76 to 80), probably because of differences in the annotation process (Table 3).

To establish the phylogeny of these phages, a database of 97 phage genomes was set up, including the genomes of S144, its two closest relatives identified in GenBank, the 93 representatives for each genus infecting *Enterobacteriaceae* and the human herpesvirus virus as outgroup. Phage S144 clearly falls in the new accepted *Loughboroughvirus* genus (Figure 4). The assignment of phage S144 to this genus was also confirmed with the alignment of four highly conserved proteins (DNA polymerase, major capsid, large terminase subunit and portal protein) from all the phages in our database where the genes were annotated (Figure A6). The closest clade to the *Loughboroughvirus* genus is another new genus, the *Rosemountvirus*, represented by the *Salmonella* phage BP63 (Figure 4). Given the high similarity (Table 3) and synteny of phage genomes in these two genera (Figure 5), we propose that they should form a new subfamily, the ‘Salusvirinae’, called after the latin for “health”, since it groups phages for food safety and it is the name of the faculty where phage S144 was isolated. Phylogenetically, the closest phage to the new proposed subfamily is phage 9g, a *Siphoviridae* infecting *E. coli* [48], both considering the nucleotide sequence of the whole genome (Figure 4) and the amino acid sequence of the DNA polymerase and the large subunit of terminase (Figure A6). In contrast, the major capsid and the portal protein identify the closest relatives as *Siphoviridae* phages (*Salmonella* phage Jersey and *E. coli* phages K1G and HK578; Figure A6), thus confirming the highly mosaic nature of S144 and the ’Salusvirinae’phages.

The proximity of phage 9g in the phylogenetic tree based on the terminase suggests that ´Salusvirinae’ phages may also have direct terminal repeats too. To establish the DNA packaging system of phage S144, the DNA was digested with two restrictions enzymes (SspI, PacI) and the combination of the two, denatured and cooled down fast or slowly (Figure A4a). The restriction pattern is not coherent with a circular form, since additional bands are present (Figure A4c). In the PacI restriction pattern, the band between 7000 bp and 10,000 bp indicates the presence of a physical end 9000 bp before (in cluster C) or after (in the gap between cluster E and A) the PacI restriction site (light blue dots in Figure A4b). Furthermore, in the SspI restriction pattern, two of the bands between 5000 bp and 7000 bp do not correspond to the expected (red dots in Figure A4a), but they substitute the expected band at 12,000 bp. In the restriction pattern from the two enzymes together, the two of the bands between 5000 bp and 7000 bp are still visible and additional band at 500 bp is substituting the expected band at 12,000 bp, thus confirming that the end is not in the cluster C, but in the gap between cluster E and A. Since no difference in the pattern was observed after cooling the restricted samples (SspI, PacI or the combination of the two) fast or slowly, it is possible to exclude cohesive ends [49]. Our data support the hypothesis of short direct repeat ends between cluster E and A [49].

### 2.7. The ´Salusvirinae’ Phages Are Highly Similar Except for the Putative Receptor Binding Protein

Sequence identity within the proposed subfamily is extremely high, as clearly visible in the whole-genome pairwise comparison matrix (Figure 5A), where the identity ranges from 65% to 96.46% at the nucleotide level. The variability within the subfamily is focused in few genes that vary particularly between the two genera: the genes coding for the putative methyltransferase (ORF63) and the following hypothetical proteins in the C cluster (ORFs 64–65), hypothetical proteins in the E (ORFs 76–78) and in the A cluster (ORFs 02, 05, 06) as well as the tail fibers and associated chaperones (ORFs 31–33) (Figure 5B). As described, in phage S144 the tail fiber is presumably encoded by ORF31 and shows structural similarity to the tail fiber of phage Mu and an overall similarity in the C-terminus with the tail fibers of the *Cronobacter* phages GAP31 and GAP32. Exploiting the high synteny of the phages belonging to the *Loughboroughvirus* genus to S144, we could compare and observe within each genus high identity in the sequence of the putative tail fiber (from 96.62% to 99.77% within the *Rosemountvirus* and from 70.42% to 99.78% within *Loughboroughvirus*). Interestingly, the C-terminus of the tail fiber in the *Loughboroughvirus* genus is more similar to the *Cronobacter* phages GAP31 and GAP32 than the *Rosemountvirus*. Since the C-terminus is the tail fiber part involved in the host recognition and binding, we suggest that also SE4 and ZCSE2 may infect *C. sakazakii* and be polyvalent phages. The infectivity could be instead very different for the phages in the *Rosemountvirus* genus, given the differences in the C-terminus of the tail fibers observed.

## 3. Discussion

Polyvalent phages are able to infect and produce progeny in at least two different bacterial host genera [50]. Although rare, polyvalent phages have been isolated using classic methods and their broad host range has been discovered by challenging them against strains of other genera than their isolation host. The same approach was used here to demonstrate that phage S144 isolated on *Salmonella* Infantis is a polyvalent myovirus, infecting *Salmonella* spp. as well as *C. sakazakii*, two well-distinct members of the *Enterobacteriaceae* genera [20,21]. Analysis of the genomic DNA revealed that the phage DNA is modified on the cytosine and encodes 80 genes in five functional modules (A, B, C, D and E). From an applied perspective, it is important to note that phage S144 does not encode any antibiotic resistance genes, virulence factors, or integrases, thus indicating a strictly lytic lifestyle that makes phage S144 a suitable candidate for phage therapy and biocontrol targeting different genera [51].

A detailed analysis of the genome revealed that phage S144 uses both self and host enzymes for key processes during its life cycle. Our bioinformatic analysis shows that S144 encodes its own DNA polymerase, in principle allowing the phage to replicate its DNA independently from the hosts, while it may use host cofactors [4]. As in other phages, the DNA replication genes are located in close proximity [52] and for phage S144 they are primarily encoded in cluster C. Moreover, S144 is able to produce pyrimidines for its DNA synthesis, as it encodes five genes predicted to be involved in the synthesis of thymidine triphosphate (dTTP). Other phages also encode genes involved in pyrimidine metabolism such as *Bacillus subtilis* phage PBS1 [53] and *Ralstonia* phage PhiRSL1 [54]. Interestingly, phage T4 encodes a tightly regulated and almost complete nucleotide biosynthesis pathway that reflects the low GC content of T4 DNA [55,56]. Similarly, the S144 genes encoding for the pyrimidine pathway may allow the phage to synthetize a genome with a lower GC content (45.8%) compared to its hosts, *S. enterica* (approximatively 52%) [57] and *C. sakazakii* (approximatively 57%) [58]. When it comes to transcription, phage S144 uses the host RNA polymerase to initiate transcription of its own genes, like phages T4 and Lambda do [59]. In accordance with this, we identified several promoter sequences highly similar to host promoters (both *Salmonella* spp. and *Cronobacter* spp., Appendix A) and a DksA C4-type domain-containing protein that phage S144 might use to direct the RNA polymerase to its own specific promoters [28,29]. Likewise, during translation, S144 exploits the host’ tRNA pool, since no tRNA synthesis genes has been detected in its genome and the predicted compatibility of phage codon usage with the hosts’ tRNA pools. Given the correlation between codon usage and tRNA [60], these data might indicate a certain degree of adaptation of S144 to its hosts. This would be a key adaptation considering that at least one of the two hosts, *Salmonella*, is translationally biased, i.e., it preferentially uses a subset of codons and their tRNA [61]. Additional experimental data on the fitness effects of varying the amino acid frequencies are necessary to confirm this hypothesis.

Polyvalent phages are the result of a sophisticated evolution process, as every single step of the infection has to be compatible with multiple hosts. So far, the polyvalent phages described in literature infect diverse hosts by recognizing a receptor that is conserved across genera, such as PrD1, that infect *Enterobacteriaceae* and *P. aeruginosa*, binding a type IV trans-envelope DNA translocation complex on an IncP plasmid [15]. Alternatively, they are equipped with multiple RBPs, as in the case of phage Phi92 and CBA120, which infect different *Enterobacteriaceae* genera [13,16]. Here, we show that, most probably, the O-antigen is the receptor for phage S144, since it is needed for infection on both *S.* Muenster and *C. sakazakii,* but our data do not exclude that a second receptor might also be involved (for example fimbriae for *S*. Muenster or capsules for *C. sakazakii*). While the two genera differ in their O-antigen structure, they do share sugar residues (a rhamnose and a galactose) that may act as a common receptor for phage S144. Some variation of the O-antigen receptor is tolerated by some phages, such as the podovirus HK620 that cleaves the O-antigen of *E. coli* belonging to serogroup O18A both in the absence or presence of a branching glucose [62]. However, further experiments are needed to determine if O-antigen is the only receptor needed for S144 to eject its DNA in *S.* Muenster and *C. sakazakii*, as seen for phages HK620 and P22 [62,63], or if a secondary receptor is required for infection, as reported for phage T4 [64].

Receptors are a major determinant of the phage host range [6]. The specificity of the interaction relies on the RBP structure, located on the tip of the tail spikes or fibers, and primarily determining which hosts the phage can bind to. Here, we propose that the tail fiber of phage S144 is encoded by ORF31, located upstream of two very similar putative chaperones. Previously, the homologous of ORF09 of the highly related phage SE4 was proposed to encode an additional tail fiber [35]. However, the structure prediction of the analogous gene in S144 (ORF17) using PHYRE2 did not resemble a tail spike or a tail fiber (data not shown) and ORF17 is thus annotated as a tail protein, even though HHPRED predicts that it might have the hydrolase activity seen in some tail spikes [36]. Our evidence of ORF31 as being the putative tail fiber is based on the level of structural similarity to tail fiber of bacteriophage Mu G+. A recent crystal structure showed that the tail fiber and the chaperone of Mu form a complex and that both proteins are part of the mature virion [37]. However, none of the two chaperones of S144 were identified in the proteome analysis, suggesting that they are not part of the mature virion. The need for two chaperones to fold the tail fiber has been previously observed in T4, where it has been suggested that gp57A might avoid the unspecific aggregation of monomers, while gp38 could initiate the folding [65]. Considering that the two fiber chaperones in phage S144 are very similar, we do not expect such as a different role as for gp57A and gp38, but further research is needed to verify this hypothesis.

Taxonomically, phage S144 can be classified as a member of the recently approved *Loughboroughvirus* genus, containing only two other members: the *Salmonella* phages ZCSE2 [43] and SE4 [35]. Since these phages are highly similar to the members of the *Rosemountvirus* genus, we proposed a new subfamily, the ‘Salusvirinae’. Phylogenetic analysis of the whole genome and of four well conserved proteins from other 97 *Enterobacteriaceae* phages shown few variations in the phages neighboring the *Loughboroughvirus* genus and the ‘Salusvirinae’ subfamily. These might result from recombination events between different phages [66,67]. The eight phages analyzed (three from the *Loughboroughvirus* genus and five from the *Rosemountvirus* genus) have a GC content of 45–46%, a genome size between 50936 and 54894 bp, no tRNA synthesis genes, high synteny and similarity ranging from 65% to 94 %.

Despite the high genome similarity, the two genera within the proposed ‘Salusvirinae’ subfamily differ in specific genes, included the tail fiber genes. Tail fiber genes are the most variable part of a phage genome and they often have a mosaic structure, especially in the C-terminal domain that determines the recognition of the receptor and the specificity of the phage [68,69,70]. Here, we found that the tail fibers of the *Loughboroughvirus* genus share a common C-terminus that is different from the *Rosemountvirus* genus, suggesting that the genera most likely differ in their host range. This was confirmed by the host range data published in the study of phages SE4 and SE13, belonging to the two genera [35]. On a panel of 61 *Salmonella* strains belonging to 34 different serovars, SE4 had a narrower host range (infecting ten strains less) than phage SE13 [35]. While these phages were tested in the same study, the extreme diversity of *Salmonella* serovars and strains makes it difficult to compare host ranges from different studies, especially since phage sensitivity may even be strain dependent. For example, phage LSE7621 (*Rosemountvirus*) was suggested to have a narrow host range, only infecting *Salmonella* Enteritidis [47]. However, LSE7621 was only tested on few strains from each of the 12 serovars included and may thus be able to infect a broader range of *Salmonella* serotypes, if tested on the same 34 *Salmonella* serovars used for phage SE13 [35,47]. Yet, the host range of PA13076 [46] and SE13 [35] seems to confirm that the *Rosemountvirus* phages are able to infect many different *Salmonella* serovars, but no other genera. In contrast, we have shown here that phage S144 is able to infect *C. sakazakii,* in addition to specific *Salmonella* serovars. Interestingly, the C-terminus of the tail fiber of the *Loughboroughvirus* phages (S144, SE4 and ZCSE2) is more similar to the otherwise unrelated *Cronobacter* phages GAP31 and GAP32 [71,72] than to the *Rosemountvirus* phages. While phages SE4 and ZCSE2 have been proven to infect many different *Salmonella* serovars, no bacterial strains from other genera were tested [35,43]. Given the similarity to phage S144 including the C-terminus of the tail fiber, involved in the host recognition and binding, we suggest that also phages SE4 and ZCSE2 may infect *C. sakazakii* and may thus be polyvalent phages. In summary, we propose that the new sub-family ‘Salusvirinae’ contains both a genus of broad host range phages infecting diverse *Salmonella* serovars and a genus of polyvalent phages, infecting both *Salmonella* species and *C. sakazakii*, and that their host range could be predicted by the C-terminus of the tail fiber.

## 4. Materials and Methods

### 4.1. Bacterial Strains

Three strains have been used for propagation of phage S144: S15 (*Salmonella* Infantis, also isolation strain; [6]), S394 (*Salmonella* Muenster) and Cs2730 (*Cronobacter sakazakii*). A total of 211 *Enterobacteriaceae* strains have been tested for sensitivity to the S144 phage (Table 1), including among others the most prevalent *Salmonella* serotypes isolated from pork meat between 2011 and 2015 (Table A1), the ECOR collection representing the diversity of *E. coli* [73] and 14 *Cronobacter sakazakii* strains kindly provided by Arla Food.

### 4.2. Phage Propagation and Plaque Assay

For preparing bacterial lawns, 100 or 300 μL overnight cultures of the selected propagation strain grown in LB (Lysogeny Broth, Merck, Darmstadt, Germany) at 37 °C were mixed with 3.3 or 11 mL of molten overlay agar (LBov; LB broth with 0.6% Agar bacteriological no.1, Oxoid) and spread on 9 or 12 cm LA (LB with 1.2% agar) plates, respectively. After settling for 5 min, lawns were dried in a laminar hood for 35 min and used immediately thereafter. To determine phage titers and host ranges, tenfold serial dilutions (up to 10^−7^−10^−9^) of the phage stocks in SM buffer (0.1 M NaCl, 8 mM MgSO4.7H2O, 50 mM Tris-HCl, pH 7.5), were prepared and 10 μL aliquots were spotted on bacterial lawns. After incubation, plaques were counted and plaque forming units per ml (pfu/mL) were calculated for each strain. Plaque assays were done for at least two independent replicates and if plaques formed in at least one of the assays, the log pfu/mL was noted. Phage stocks were prepared by plate lysis method [74].

### 4.3. Transmission Electron Micrographs

A high titre suspension of phage S144 was sedimented for 60 min at 25,000× *g*. Supernatant was replaced with 0.1 M ammonium acetate solution (pH 7) and re-centrifuged. After three rounds of washing and subsequent 30-min fixation with 1% (*v*/*v*) glutaraldehyde (EM-grade), final phage suspension was stained with 2% (*w*/*v*) uranyl acetate on freshly prepared carbon films. Grids were analysed in a Tecnai 10 TEM (FEI Thermo Fisher Scientific Company, Eindhoven, the Netherlands) at an acceleration voltage of 80 kV. Micrographs were taken with a MegaView II charge-coupled device camera (Emsis, Muenster, Germany). Capsid, tail and tail fibers dimensions were measured on at least 15 phage particles.

### 4.4. Identification of the Receptor

To isolate mutants resistant to the phage S144, two lawns with 10^5^ pfu/mL of S144 propagated on the wild type strains *S*. Muenster S394 and *C. sakazakii* CS1 have been used as surface to streak the same two wild type strains. Colonies growing on phage lawn has been picked and further streaked other three times on lawns with 10^5^ pfu/mL of S144. At each step, the resistant colony was also saved as frozen culture and tested to confirm its sensitivity to S144 by spot assay. For each strain, one mutant (from the last frozen stocks) and the wild type were used to prepare an overnight culture for DNA extraction with DNeasy Blood & Tissue kit (Qiagen, Hilden, Germany) and sequenced with MiSeq (Illumina, San Diego, CA, USA). Reads from wild type strains were de novo assembled using CLC Genomics Workbench 9.5.3 (Qiagen, Aarhus, Denmark), the contigs were RAST annotated and regions coding for the lipopolysaccharide were manually curated, according to what reported for the O-antigen of *S*. Muenster [23] and *C. sakazakii* CS1 [22]. Reads from mutants were mapped against the wild type sequence and gaps (with coverage lower than 5 reads) were identified using CLC Genomics Workbench 9.5.3 (Qiagen, Aarhus, Denmark) and are summarized in Excel files available as Table A3 and Table A4.

### 4.5. Adsorption Assay

Adsorption of phage S144 to the wild type and resistant strains of *S.* Muenster S394 and *C. sakazakii* CS1 was determined as in [75]. The concentrations of bacteria and phages used in the assay were specifically adjusted according to the different sensitivity of the two strains to S144. For *S.* Muenster S394, an overnight culture (LB, 37 °C, 1800 rpm) was used to inoculate two tubes with 40 mL LB each, to an optical density (OD_600_) of 0.6, corresponding to 4 × 10^8^ cfu/mL (colony forming units/mL). Only sterile SM buffer (0.4 mL) was added to the control tube, while 0.4 mL of 4 × 10^8^ pfu/mL from the phage stock was added to the test tube to obtain a multiplicity of infection (MOI) of 0.01. For *C. sakazakii* CS1, an overnight culture (LB, 37 °C, 1,800 rpm) was used to inoculate two tubes with 40 mL LB each, to OD_600_ of 0.6, corresponding to 4 × 10^8^ cfu/mL. Only sterile SM buffer (0.4 mL) was added to the control tube, while 0.4 mL of a 1 × 10^7^ pfu/mL phage stock was added to the test tube in order to obtain a MOI of 0.001. All tubes were incubated at 37 °C, 1800 rpm, and, after 0, 10, 20 and 60 min from the phage addition, 0.7 mL from both tubes were collected in a syringe, filtered through 0.2-μm-pore size filters (Thermo Fisher Scientific, Waltham, MA, USA), diluted and spotted on a lawn of the wild type strain for counting. The concentration of free phages was calculated from the phages not adsorbed in each sample. Experiments were performed in triplicate. Statistical analysis of variance (ANOVA) and Tukey’s test were applied to evaluate any significant differences among the samples (*p*-values < 0.05).

### 4.6. Phage DNA Extraction and Sequencing

High titer phage stocks (10^9^ pfu/mL) were subjected to phenol-chloroform based DNA extraction and purification by ethanol precipitation with modifications [76]. Briefly, RNAse (Thermo Fisher Scientific, Waltham, MA, USA) and DNAse (Thermo Fisher Scientific) were added to phage stocks to final concentrations of 10 and 20 μg/mL, respectively, and left for digestions at 37 °C for 2–3 h in a thermo-shaker (500 r.p.m., Eppendorf, Hamburg, Germany). Following addition of EDTA (20 mM) and proteinase K (50 μg/mL, Thermo Fisher Scientific) and incubation at 56 °C for 4 h, phenol (Fluka, Buchs, Switzerland), phenol-chloroform-isoamylalcohol (25:24:1, Ambion, Austin, TX, USA) and three rounds of chloroform-isoamylalcohol (24:1) treatment were performed. To precipitate the DNA, 0.1 volume of 3 M sodium acetate (pH 5.5), glycogen (final concentration of 0.05 μg/μL, Thermo Fisher Scientific) and 2.5 volume of ice-cold ethanol (99.9%) were added. After incubation at −20 °C for up to 72 h, precipitated DNA was centrifuged at 31,000× *g* for 20 min, washed three times with 70% ice cold ethanol and dissolved in 10 mM Tris–HCl (pH 8.0). DNA concentrations were measured using Qubit (Thermo Fisher Scientific). DNA libraries were prepared using Nextera XT v.3 (Illumina, San Diego, CA, USA) kit. Next generation sequencing was performed using MiSeq (Illumina) platform with paired-end (2 × 250-bp) operating mode.

### 4.7. Genome Assembly and Bioinformatic Analysis

Sequencing reads were assembled de novo using CLC Genomics Workbench 9.5.3 (Qiagen, Aarhus, Denmark). A consensus sequence was obtained with a minimum of 30-fold coverage. Analysis and annotation of the phage genome were performed using tools in Galaxy (https://cpt.tamu.edu/galaxy -pub) and Web Apollo, hosted by the Center for Phage Technology at Texas A&M University (CPT Galaxy) [77,78]. Gene calling was performed using GLIMMER 3.0 and MetaGeneAnnotator 1.0 within the structural workflow, while for the genome annotation the functional workflow was used, by interrogating the databases of UniProtKB Swiss-Prot/TrEMBL, Canonical Phages and HHMER. Promoters were predicted with PhagePromoter [79], selecting *Salmonella* or *Cronobacter* as hosts and manually curated. The promoter host and score calculated by PhagePromoter are available in the full annnotation table, available as Appendix A. Putative spanins have been identified with the spanin tools in CPT Galaxy and curated as suggested by Kongari [58,80]. The genome was visualised with DNA Plotter [81]. The presence of encoded tRNA genes in the phage and bacterial genomes (*P. aeruginosa* PAO1 complete genome NZ_CP053028 used as a non-*Enterobacteriaceae* control strain; *S.* Muenster whole genome NZ_CP019201 used as a positive control strain; *S.* Muenster S394; *C. sakazakii* CS1) was checked using Aragorn [82] and the phage codon usage was established with The Sequence Manipulation Suite [83]. The genomic sequence of bacteriophage S144 was deposited in the GenBank database under accession number MT663719. In addition, homolog detection and structural prediction by HHPRED [84], SWISS-MODEL [85] and, to some extent, PHYRE2 [86] were used to further investigate the function of the predicted structural proteins in S144. ThreaDomEx [87] was used for domain prediction in the putative tail fiber (ORF31). The sequence alignment and structural superposition of tail fibers of S144 and of Mu G+ (gpS) were conducted in CLC Main Workbench 20 and the structures were visualized in Pymol (version 2.3, DeLano Scientific, San Carlos, CA, USA).

### 4.8. Proteome Analysis Using ESI-MS/MS

Phage virion proteins were extracted from phages purified with polyethylene glycol (10^10^ pfu/mL) using chloroform-water-methanol extraction (1:1:0.75, *v*/*v*/*v*). The resulting protein pellet was resuspended in 20 µL of 10 mM Tris-HCl pH 6.8 and trypsinized using a gel-free method. For this, 10 µL of protein sample was mixed with 25 µL denaturation buffer (50 mM Tris-HCl pH 8.5, 6 M urea, 8.5 mM DTT) and incubated for 1 h at 56 °C in a water bath. After adding 25 µL 100 mM iodoacetamide (Sigma Aldrich, St. Louis, MO, USA) in 50 mM NH4HCO3 and 150 µL 50 mM NH4HCO3, the samples were incubated for 45 min in the dark, followed by adding 0.8 µg trypsin (Promega, Madison, WI, USA) to the samples and incubating overnight at 37 °C. Mass spectrometry analysis was performed on an Easy-nLC 1000 liquid chromatograph, coupled to a mass calibrated LTQ-Orbitrap Velos Pro via a Nanospray Flex ion source (all Thermo Fisher Scientific) using sleeved 30 µm ID stainless steel emitters. Peptides were identified with SEQUEST v1.4 (Thermo Fisher Scientific) and Mascot v2.5 (Matrix Sciences) and a database containing all possible translated open reading frames (ORFs), as identified using ORF finder (http://www.ncbi.nlm.nih.gov/gorf/gorf.html), and compared with the GenBank nr (non-redundant) protein database using the BLASTP.

### 4.9. Restriction Enzyme Analysis

To establish the packaging system of phage S144, the DNA from phage S144 was completely digested (37 °C for 40 min) with SspI, PacI or the two enzymes together, denatured for 15 min at 80 °C and then cooled fast (on ice) or slowly to room temperature. The products were analyzed by 0.8% agarose gel electrophoresis.

Restriction analysis was performed on S144 DNA extracted from three stocks, propagated on S15, S394 and Cs2730. S144 DNA was digested with the FastDigest restriction enzymes EheI, MauBI, PacI, Cfr42I, XhoI, NotI, Bsp120I, SspI and SspDI (respectively, FD0443, FD2084, FD2204, ER0201, FD0694, FD0596, FD0134, FD0774, FD0774 and ER2191 from Thermo Fisher). Methylated and unmethylated DNA from phage Lambda (respectively, SD0011 and SD0021) was used as control. The reaction mix was prepared with 300 ng of DNA (1 μL), 10× Fast-Digest Green Buffer (2 μL), Fast Digest enzyme (1 μL) and H_2_O nuclease free (16 μL), incubated at 37 °C in a heat block for 20 min and inactivated according to the manufacturer’s instructions (EheI and MauBI: 65 °C for 5 min, PacI: 65 °C for 10 min, Cfr42I: 65 °C for 20 min, XhoI and NotI: 80 °C for 5 min, Bsp120I: 80 °C for 10 min, SspI and SspDI: 80 °C for 20 min).

### 4.10. Taxonomy

To get an insight in S144 taxonomy, an overall genome BLASTn search was run (June 2020) and other two phages belonging to the *Loughboroughvirus* genus have been identified as closest relatives. Easyfig [88] was used to compare the S144 genome and its closest relative phage genomes and to visualise the coding regions for each phage. To define the taxonomical position of S144 and its two closest relatives within the classification of International Committee on Taxonomy of Viruses (ICTV), a phylogenetic analysis was run with the Virus Classification and Tree Building Online Resource (VICTOR) [89] with the settings recommended for prokaryotic virus nucleotide sequence (d0). In the analysis, we also included 93 phage genomes representative for each genus infecting *Enterobacteriaceae* strains and the genome of the human herpesvirus (NC_006273) as outgroup. The phylogenetic trees were visualized with iTOL [90]. In iTOL, for each phage, we indicated the family according to ICTV [91].

For confirmation, phylogenetic analyses of the amino acid sequence of DNA polymerase, major capsid, large terminase subunit and portal protein were carried out. For each tree, we aligned the amino acid translation of the coding sequence from S144 with the ones from the two closest relatives and from the 94 phages representative for each genus infecting *Enterobacteriaceae* strains, where the genes were clearly annotated. Multiple sequence alignments of phage orthologue proteins and generation of phylogenetic trees were performed using the Maximum Likelihood Phylogeny 1.2 from CLC Workbench with default parameters (construction method: neighbour joining; protein substitution model: WAG) and visualized with iTOL [90].

## Figures and Tables

**Figure 1 ijms-21-05196-f001:**
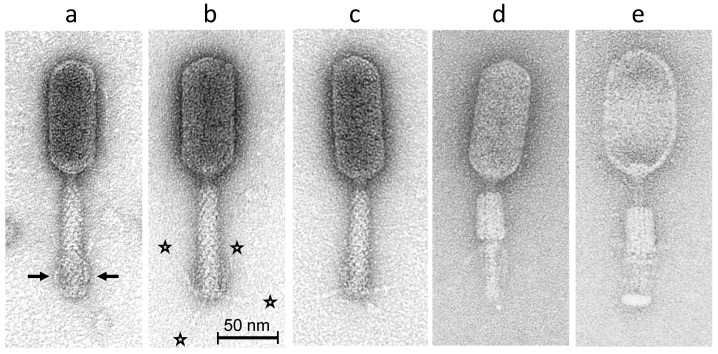
Transmission electron micrographs of phage S144 negatively stained with 2% (*w*/*v*) uranyl acetate. Three intact phage particles with extended tail sheaths are shown in (**a**–**c**) with short and notably rigid tail fibers attached to the distal tail region in upward position (i.e., in retracted configuration indicated by arrows in (**a**) or in (randomly) extended configurations ((**b**,**c**); see asterisks in (**b**)). Phage particles with contracted tail sheaths are shown in (**d**) (with intact capsid) and in (**e**) (with empty capsid). Bar represents 50 nm, as indicated.

**Figure 2 ijms-21-05196-f002:**
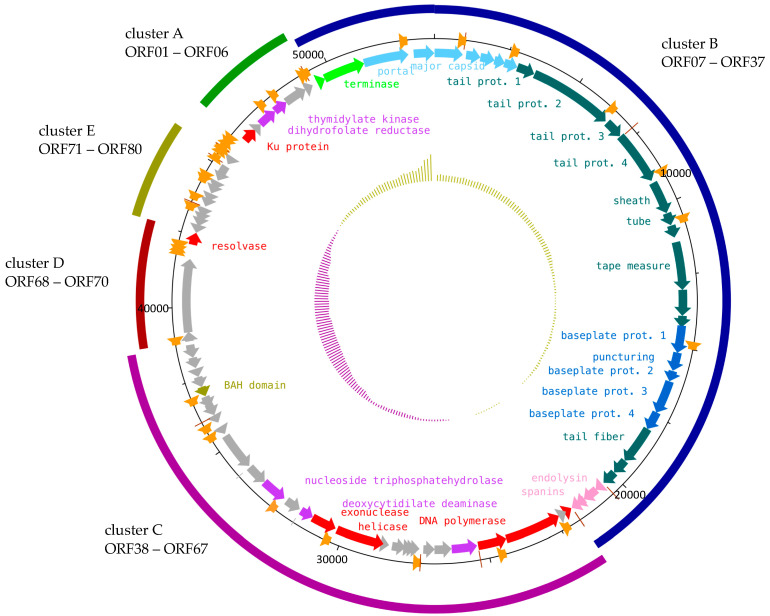
Genomic organization and functional modules of phage S144. The inner circle shows the GC content (below average in purple, above average in ocher). Putative genes for phage morphogenesis are indicated as blue arrows (light blue: capsid, cyan: tail, blue: baseplate), genes for DNA packaging in green, lysis-associated genes in pink, genes involved in DNA manipulation in red, genes for nucleotide biosynthesis in purple and additional functions in ocher (prot. = protein). The middle circle indicates the promoters (orange arrows) and terminators (brown lines). Asterisks indicate the gene products identified in the proteomic analysis while the external ring highlights the identified clusters (from **A** to **E**).

**Figure 3 ijms-21-05196-f003:**
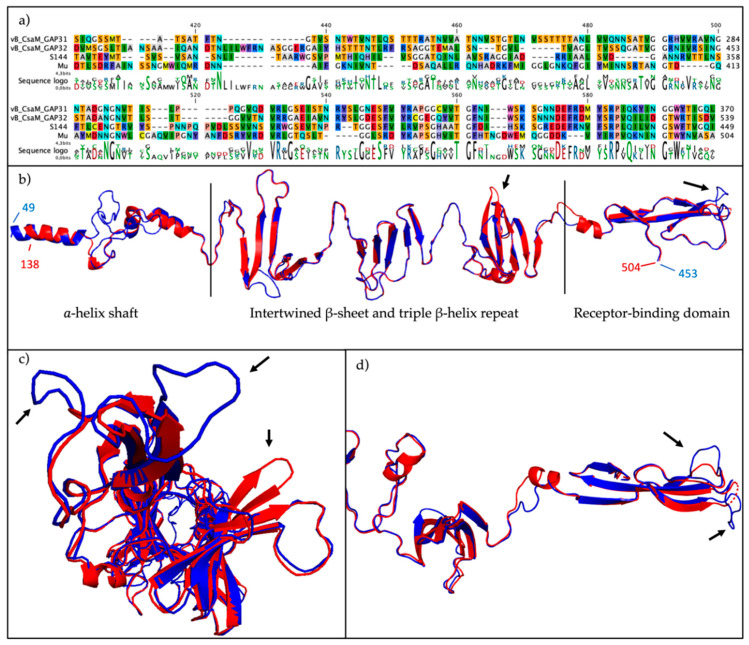
Sequence and structural alignment of the predicted S144 tail fiber protein. (**a**) Sequence alignment of the C-terminal residues of the tail fibers from phages S144, GAP31, GAP32 and Mu G+. (**b**) Structural comparison by superposition of the experimental resolved tail fiber protein from phage Mu G+ (red; PDBID:5YVQ) and the predicted S144 tail fiber (blue) (RMSD 1,2 Å over 348 matched residues, TM-score 0.843) with the three domains determined in Mu’s tail fiber, (**c**) from the distal end and (**d**) zoom in of the C-terminus, involved in the cell surface binding. The arrows indicate structural differences between the two tail fibers.

**Figure 4 ijms-21-05196-f004:**
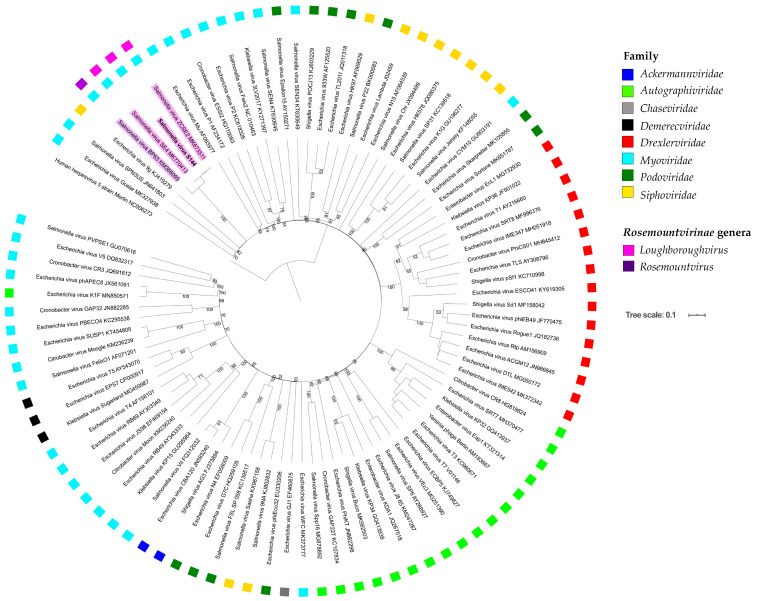
Phylogenomic analysis at the nucleotide level of phage S144, its two closest relatives, and 93 phage genomes representative for each genus infecting *Enterobacteriaceae* strains and the genome of the human herpesvirus as outgroup. The tree has been built using VICTOR with the formula d0, recommended for phages, and aesthetically modified with iTOL. Bootstrap values higher than 50% are indicated on the branches. The branch lengths of the resulting VICTOR trees are scaled in terms of the used distance formula. S144 and the other two phages in the *Loughboroughvirus* virus are highlighted in pink and the representative from the closest genus, the *Rosemountvirus* genus, in purple. For each phage, we indicated the accession number and the family, according to ICTV.

**Figure 5 ijms-21-05196-f005:**
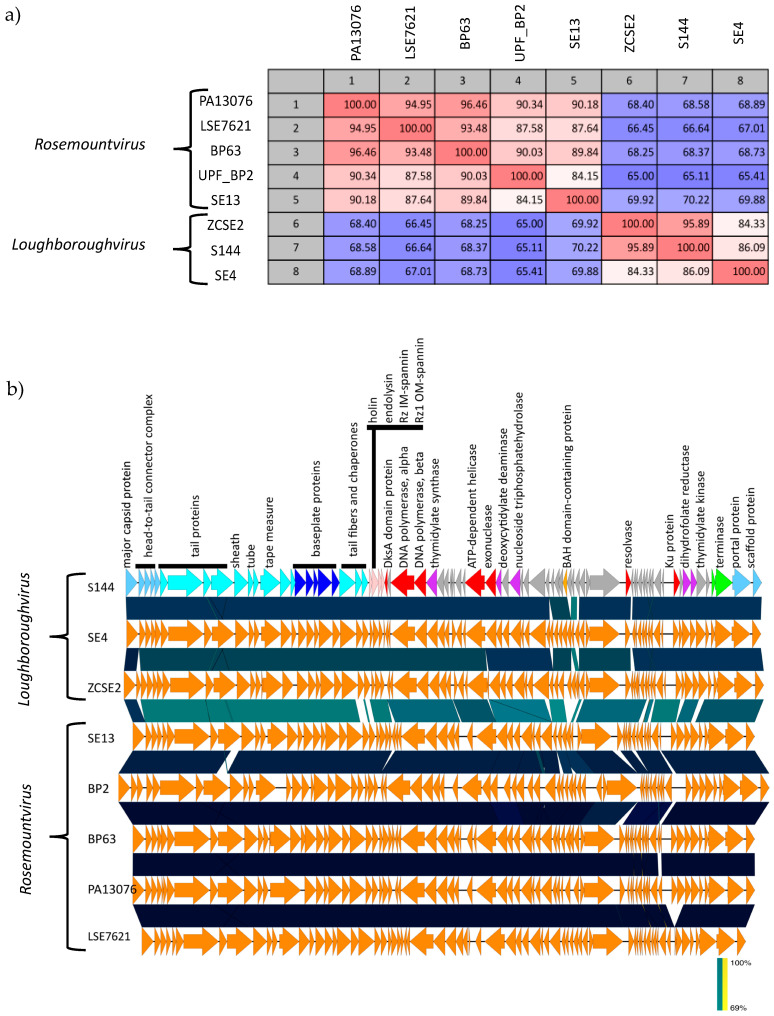
Genomic comparison at the nucleotide level of S144 genome to the other two phages in the *Loughboroughvirus* genus and five representatives from the closest genus, the *Rosemountvirus* genus. (**a**) Whole-genome pairwise comparison of the eight phage nucleotide sequences, visualized as a matrix with percent identity with CLC Main Workbench 7. (**b**) Organization and functional modules of the eight phage genomes. This figure was generated using EasyFig. Genes for phage morphogenesis are marked in blue (light blue: capsid, cyan: tail, blue: baseplate), genes for DNA packaging in green, putative genes for lysis in pink, genes involved in DNA manipulation in red, putative genes for nucleotide biosynthesis in purple and additional functions in ochre (prot. = protein). Blue shade indicates the level of similarity.

**Table 1 ijms-21-05196-t001:** Summary of the host range of phage S144, with genus, species or serovar and number of infected over the tested strains for each bacterial taxon (positive infections in orange). The full host range is reported in the Table A1 with the corresponding efficiency of plating (EOP) for each strain.

Genus	Species or Serovar	Infected/Tested Strains	Reference
*Salmonella*	Derby	2/21	[6]
*Salmonella*	Typhimurium	0/13	[6]
*Salmonella*	Dublin	0/4	[6]
*Salmonella*	Enteritidis	2/3	[6]
*Salmonella*	Infantis	3/4	[6]
*Salmonella*	Rough	0/4	[6]
*Salmonella*	4.12:i:-	0/6	[6]
*Salmonella*	4.5.12:i:-	1/9	[6]
*Salmonella*	4.5.12:i:-	1/1	[6]
*Salmonella*	Goettingen	1/1	[6]
*Salmonella*	Livingstone	0/1	[6]
*Salmonella*	London	1/1	[6]
*Salmonella*	Rissen	0/1	[6]
*Salmonella*	Brandenburg	0/1	[6]
*Salmonella*	Bradford	0/1	[6]
*Salmonella*	Senftenberg	0/1	this work
*Salmonella*	Adelaide	0/1	this work
*Salmonella*	Weslaco	0/1	this work
*Salmonella*	Montevideo	0/1	this work
*Salmonella*	Tanger	1/1	this work
*Salmonella*	Cerro	0/1	this work
*Salmonella*	Basel	0/1	this work
*Salmonella*	Anatum	0/1	this work
*Salmonella*	Eilbek	1/1	this work
*Salmonella*	Worthington	0/1	this work
*Salmonella*	Onderstepoort	1/1	this work
*Salmonella*	Deversoir	1/1	this work
*Salmonella*	Telaviv	1/1	this work
*Salmonella*	Choleraesuis	1/1	this work
*Salmonella*	Aberdeen	1/1	this work
*Salmonella*	Inverness	1/1	this work
*Salmonella*	Bergen	0/1	this work
*Salmonella*	Ruiru	0/1	this work
*Salmonella*	Gaminara	1/1	this work
*Salmonella*	Paratyphi B var. Java	0/1	this work
*Salmonella*	Muenster	1/1	this work
*Escherichia*	*coli*	0/77	this work
*Klebsiella*	*spp.*	0/3	this work
*Enterobacter*	*spp.*	1/4	this work
*Providencia*	*stuarti*	0/1	this work
*Citrobacter*	*spp.*	0/2	this work
*Yersinia*	*spp.*	0/1	this work
*Serratia*	*marcescens*	0/1	this work
*Erwinia*	*herbicola*	0/1	this work
*Shigella*	*sonnei*	0/1	this work
*Acinetobacter*	*spp.*	0/1	this work
*Proteus*	*mirabilis*	0/2	this work
*Cronobacter*	*sakazakii*	4/15	this work

**Table 2 ijms-21-05196-t002:** Open reading frames (ORFs) coding for structural proteins in the genome of phage S144: their putative functions, functional category, assigned cluster; protein molecular weight in KDa, number of unique peptides and sequence coverage. The full annotation table with notes on similarities and homologies found for each predicted ORF in the databases is reported in the Appendix A. Abbreviations: put. = putative; prot. = protein; hyp. = hypothetical.

ORF	Product	Functional Category	Cluster	Proteins			
				gp	MW (kDa)	Unique Peptides	Coverage (%)
ORF03	put. dihydrofolate reductase	nt biosynthesis	A	gp03	25	3	25
ORF08	terminase, large subunit	packaging	B	gp08	54	2	4
ORF09	portal prot.	capsid morphogenesis	B	gp09	58	12	27
ORF10	scaffold prot.	capsid morphogenesis	B	gp10	27	2	5
ORF11	major capsid prot.	capsid morphogenesis	B	gp11	35	14	49
ORF12	put. head-to-tail connector complex 1	capsid morphogenesis	B	gp12	18	10	49
ORF14	put. head-to-tail connector complex 3	capsid morphogenesis	B	gp14	14	5	52
ORF15	put. head-to-tail connector complex 4	capsid morphogenesis	B	gp15	17	3	22
ORF16	put. tail prot. 1	tail morphogenesis	B	gp16	21	3	24
ORF17	put. tail prot. 2	tail morphogenesis	B	gp17	111	25	33
ORF18	put. tail prot. 3	tail morphogenesis	B	gp18	22	17	80
ORF20	put. sheath prot.	tail morphogenesis	B	gp20	42	9	32
ORF21	put. tube prot.	tail morphogenesis	B	gp21	16	6	26
ORF23	put. tape measure prot.	tail morphogenesis	B	gp23	60	10	26
ORF24	conserved hyp. prot.	tail morphogenesis	B	gp24	32	11	31
ORF25	conserved hyp. prot.	tail morphogenesis	B	gp25	14	4	27
ORF26	put. baseplate prot. 1	baseplate morphogenesis	B	gp26	35	9	32
ORF27	put. puncturing prot.	baseplate morphogenesis	B	gp27	23	6	50
ORF28	put. baseplate prot. 2	baseplate morphogenesis	B	gp28	14	5	45
ORF29	put. baseplate prot. 3	baseplate morphogenesis	B	gp29	42	10	30
ORF30	put. baseplate prot. 4	baseplate morphogenesis	B	gp30	24	7	33
ORF31	put. tail fiber prot.	tail morphogenesis	B	gp31	48	9	24
ORF35	put. endolysin	lysis	B	gp35	20	5	28
ORF36	put. IM-spanin	lysis	B	gp36	12	5	38
ORF37	put. OM-spanin	lysis	B	gp37	10	1	10
ORF40	hyp. prot.	hyp.	C	gp40			
ORF43	thymidylate synthase	nt biosynthesis	C	gp43	33	5	21
ORF54	conserved hyp. prot.	hyp.	C	gp54	19	2	21
ORF55	put. P-loop with nucleoside triphosphatehydrolase	nt biosynthesis	C	gp55	33	2	9
ORF58	conserved hyp. prot.	hyp.	C	gp58	49	5	16
ORF69	conserved hyp. prot.	hyp.	D	gp69	96	2	4
ORF79	conserved hyp. prot.	hyp.	E	gp79	21	2	14

**Table 3 ijms-21-05196-t003:** Morphological (morphotype, capsid shape) and genomic features (percentage of coding genome, GC content, genome size, ORFs, tRNAs synthesis genes) of S144, its two closest relatives belonging to the *Loughboroughvirus* genus and five representative phages from the *Rosemountvirus* genus. The GenBank accession number and the reference are also reported. ND: not determined.

Genus	Phage	Morphotype	Capsid Shape	Coding (%)	GC (%)	Genome Size (bp)	ORFs	tRNAs	GenBank Acc. No.	Ref.
*Loughboroughvirus*	*Salmonella* phage S144	myovirus	prolate, 101 × 44 nm	93.4	46	53,628	80	0	MT663719	this work
	*Salmonella* phage SE4	myovirus	ND	92.2	45	53,494	76	0	MK770413.1	[35]
	*Salmonella* phage ZCSE2	myovirus	prolate 84 × 35 nm	92.0	46	53,965	78	0	MK673511.1	[43]
*Rosemountvirus*	*Salmonella* phage BP63	myovirus	ND	93.2	46	52,437	76	0	KM366099.1	[44]
	*Salmonella* phage SE13	myovirus	ND	91.0	46	52,438	73	0	MK770411.1	[35]
	*Salmonella* phage UPF_BP2	myovirus	ND	85.2	46	54,894	70	0	KX826077.1	[45]
	*Salmonella* phage vB_SenM_PA13076	myovirus	oval Ø 66 nm	89.2	46	52,474	68	0	MF740800.1	[46]
	*Salmonella* phage LSE7621	myovirus	symmetrical Ø 56 nm	92.1	46	50,936	72	0	MK568062.1	[47]

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
