# Peer review of "Phage S144, a New Polyvalent Phage Infecting Salmonella spp. and Cronobacter sakazakii"

_ijms, 2020, doi:10.3390/ijms21155196_

Round 1

Reviewer 1 Report

The manuscript entitled “Phage S144, a new polyvalent phage infecting Salmonella spp. and Cronobacter sakazakii” by Gambino et al. reported genomic and proteomic characteristics of a novel polyvalent phage S144 which isolated from wastewater. Through a comprehensive genetic analysis of the phage-resistant mutants, the authors proposed that this phage S144 might utilize O-antigens as receptors for binding to Salmonella spp. and Cronobacter sakazakii cells. They also revealed a similarity between the putative tail fiber of S144 and that of other phages in nucleotide or protein level, and based on this, suggested the key role of tail fiber in host determination. Overall, experiments and analyses were appropriately designed and performed, and thus, it could present several key results for their suggestions. But the following points need to be addressed or corrected to be considered as a publishable article in IJMS.  

  1. There are missing refs. in the References section (i.e., Although refs. 92-96 are cited in the manuscript but absent in the References)
  2. Line 28. According to your analysis in the manuscript (line 264), S144 tail fiber is similar to that of phage Mu in overall structure, more specifically in the distal C-term, not in N-term region. What is the correct one?
  3. Line 106. According to the Table 1 etc., you used 15 C. sakazakii strains and 4 Enterobacer spp. Not 14 and 5, respectively.
  4. Line 133. You did not check the presence of O-antigens in the mutants through any analysis or experiments, but just speculated the lack of O-antigen based on the genetic analysis of the mutant. Therefore, it is more proper to be described as follows: “….and the dTDP-4-dehydrorhamnose 3,5-epimerase (rmlC) suggested that the mutant may lacks the O-antigen….”
  5. Line 146. Again, “…. O-antigen might be needed for….”
  6. Lines 167-178. Clusters in the manuscript are not match with the Fig. 2. For example, ORF71-80 are Cluster E in Fig. 2, but you indicated those are Cluster A in line 168. You need to carefully re-check the correspondence between manuscript and Figs. throughout the whole manuscript.
  7. Table 2 is too long to be contained in a manuscript. Recommend move to supplemental data, except the analyzed proteins through ESI-MS/MS.
  8. Line 288. ORF35, not ORF36.
  9. Lines 389-390. Need to be exchanged between Rosemountvirus and Loughboroughvirus.
  10. In Table A2, Log (PFU/ml), not Log (CFU/ml).
  11. Figure A1 is not suitable to see the differences between the bacterial host since individual plaques are not visible in the presented pictures. I recommend using a gel documentation system or lightbox equipped with white backlights to capture the plaques on plates, if possible.
  12. The whole manuscript needs to be carefully revised by the authors to correct many inaccuracies including above.

Author Response

Reply to Reviewer 1

We thank the Reviewer 1 for the constructive comments and suggestions. We addressed the comments point-by-point here below (reviewer’s comments in italics and our reply in bold) and we have modified our manuscript highlighting changes by using the track changes mode.

The manuscript entitled “Phage S144, a new polyvalent phage infecting Salmonella spp. and Cronobacter sakazakii” by Gambino et al. reported genomic and proteomic characteristics of a novel polyvalent phage S144 which isolated from wastewater. Through a comprehensive genetic analysis of the phage-resistant mutants, the authors proposed that this phage S144 might utilize O-antigens as receptors for binding to Salmonella spp. and Cronobacter sakazakii cells. They also revealed a similarity between the putative tail fiber of S144 and that of other phages in nucleotide or protein level, and based on this, suggested the key role of tail fiber in host determination. Overall, experiments and analyses were appropriately designed and performed, and thus, it could present several key results for their suggestions. But the following points need to be addressed or corrected to be considered as a publishable article in IJMS.

  1. There are missing refs. in the References section (i.e., Although refs. 92-96 are cited in the manuscript but absent in the References)

Answer: the missing references have been added (lines 1028-1037).

  1. Line 28. According to your analysis in the manuscript (line 264), S144 tail fiber is similar to that of phage Mu in overall structure, more specifically in the distal C-term, not in N-term region. What is the correct one?

Answer: we thank the reviewer for his/her comment. There is an overall similarity with phage Mu tail fiber and more specifically in the C-term. We corrected the mistake in line 29.

  1. Line 106. According to the Table 1 etc., you used 15 C. sakazakii strains and 4 Enterobacer spp. Not 14 and 5, respectively.

Answer: the reviewer is right. We corrected the mistakes (line 112).

  1. Line 133. You did not check the presence of O-antigens in the mutants through any analysis or experiments, but just speculated the lack of O-antigen based on the genetic analysis of the mutant. Therefore, it is more proper to be described as follows: “….and the dTDP-4-dehydrorhamnose 3,5-epimerase (rmlC) suggested that the mutant may lacks the O-antigen….”

Answer: we thank the reviewer for his/her suggestion and we changed the sentence accordingly (line 144).

  1. Line 146. Again, “…. O-antigen might be needed for….”

Answer: we changed the sentence accordingly (line 158).

  1. Lines 167-178. Clusters in the manuscript are not match with the Fig. 2. For example, ORF71-80 are Cluster E in Fig. 2, but you indicated those are Cluster A in line 168. You need to carefully re-check the correspondence between manuscript and Figs. throughout the whole manuscript.

Answer: the reviewer is right. We corrected the mistakes throughout the manuscript.

  1. Table 2 is too long to be contained in a manuscript. Recommend move to supplemental data, except the analyzed proteins through ESI-MS/MS.

Answer: we thank the reviewer for his/her suggestion. We replaced Table 2 with a smaller table containing only the protein data, since the rest of information were already present in the full annotation table loaded as Supplementary material.

  1. Line 288. ORF35, not ORF36.

Answer: we corrected the mistake (line 326).

  1. Lines 389-390. Need to be exchanged between Rosemountvirus and Loughboroughvirus.

Answer: we corrected the mistake (lines 430-431).

  1. In Table A2, Log (PFU/ml), not Log (CFU/ml).

Answer: we corrected the mistake (Table A2).

  1. Figure A1 is not suitable to see the differences between the bacterial host since individual plaques are not visible in the presented pictures. I recommend using a gel documentation system or lightbox equipped with white backlights to capture the plaques on plates, if possible.

Answer: we thank the reviewer for his/her suggestion and we changed the figure accordingly. In addition, the original pictures have been added to the Supplementary material.

  1. The whole manuscript needs to be carefully revised by the authors to correct many inaccuracies including above.

Answer: we thank the reviewer for his/her comment. The manuscript has been carefully revised to spot all inaccuracies.

Reviewer 2 Report

In this article, the authors provide a thorough experimental and genetic characterization of the phage S144. The paper reflects a large amount of work, and I generally appreciated the many details that the authors provided. That said, there were a couple of instances where I did not entirely agree with the authors’ conclusions, or where some minor correction appears to be needed. These minor concerns are detailed below:

Minor Concerns:

1. Interpreting GC content: In lines 167-170, the authors point to a cluster of genes, and highlight that the GC content (38.6+/-4.6) is significantly lower than the genome average (46.1+/-2.8) and could indicate a likely HGT event. While this is an interesting possibility, I do not think the GC content provides sufficient evidence to make this claim. In particular, when viewed in light of GC content variation as shown in Figure 2, the GC content in this particular region does not seem to be especially low. Moreover, the genome-wide pattern of GC content seems to correspond to typical patterns of “GC skew” observed in bacteria and many phages, where GC skew is often used to predicted the origin or terminus. The authors later identify the terminus as appearing right between clusters E and A, which leads me to think that this explains the GC content variation, rather than HGT.

2. The use of “codon usage”: I have two issues with the discussion of codon usage in the paper. First, at lines 214-217 and in Figure A3, the authors appear to use the term “codon usage” to refer to amino acid frequency in the genome. More typically, “codon usage” refers to the relative proportion of alternative codons for each amino acid. This makes it difficult to interpret Figure A3 and some of the results, because, as far as I could tell, the actual frequency of alternative codons is not considered. I do think the comparison of amino acid usage is still interesting, but this should be clarified.

        Second, I do not think the data, as presented, are sufficient to conclude that the potential similarity in amino acid frequency represents a true “adaptation.” To really show this from an evolutionary standpoint, the authors would need to include additional information about the fitness effects of varying the amino acid frequencies. As such, I would suggest restricting this interpretation to the Discussion. Moreover, the authors might consider presenting the data in Figure A3 as a scatterplot with a regression (maybe only choosing one representative host genome for each host species) rather than as a grouped bar plot in order to streamline the presentation and emphasize the strength of the correlation between the genomes.

3. The authors should be careful relying on methods like VICTOR for inferring a global phylogeny. Because its approach depends on a whole genome distance metric, it will ignore the potential influence of paralogs and horizontal gene transfer in reconstructing the tree. It might give reasonable potential clades for the purpose of taxonomy, but caution would be needed for deeper inference. It is difficult to assess the accuracy of this kind of tree as compared with a tree based on a collection of core genes (such as could be made using the same genes included in Figure A6).

4. It would be nice to see additional detail in the Methods on what substitution models were used in CLC for the ML trees in Figure A6.

5. Figure A1: It might be an issue with the image size within the document, but I have a hard time seeing plaques in the first two panels (the plaques in panel 3 on S394 are visible).

6. Line 163 and 214 (and elsewhere): This is a very minor point, but in several places the authors mention that the genome “contains no tRNAs,” and I would recommend clarifying to read “tRNA synthesis genes.”

7. Figure A4: The enzyme labels appear to be switched on the left side of panel A, since SspI should have more bands (and also based on the color coding of the dots as described in the caption).

8. Lines 204-206 and again in the Discussion (419-420): the authors discuss promoters that “match” the hosts and are “compatible” and refer to Supplementary material. However, I was unable to determine where in the supplement I was meant to look to find this information. I would also suggest that the authors be more specific in describing what it means to match or be compatible. Are these exact matches?

9. Minor writing concerns: Overall, I found the writing clear and easy to follow, but there are a few places where the authors might consider breaking paragraphs into smaller pieces that would be easier for the reader to digest and that would maintain greater focus (e.g. the paragraphs beginning on Line 232 and 461).

Author Response

Reply to Reviewer 2

We thank the Reviewer 2 for the constructive comments and suggestions. We addressed the comments point-by-point here below (reviewer’s comments in italics and our reply in bold) and we have modified our manuscript highlighting changes by using the track changes mode.

In this article, the authors provide a thorough experimental and genetic characterization of the phage S144. The paper reflects a large amount of work, and I generally appreciated the many details that the authors provided. That said, there were a couple of instances where I did not entirely agree with the authors’ conclusions, or where some minor correction appears to be needed. These minor concerns are detailed below:

Minor Concerns:

  1. Interpreting GC content: In lines 167-170, the authors point to a cluster of genes, and highlight that the GC content (38.6+/-4.6) is significantly lower than the genome average (46.1+/-2.8) and could indicate a likely HGT event. While this is an interesting possibility, I do not think the GC content provides sufficient evidence to make this claim. In particular, when viewed in light of GC content variation as shown in Figure 2, the GC content in this particular region does not seem to be especially low. Moreover, the genome-wide pattern of GC content seems to correspond to typical patterns of “GC skew” observed in bacteria and many phages, where GC skew is often used to predicted the origin or terminus. The authors later identify the terminus as appearing right between clusters E and A, which leads me to think that this explains the GC content variation, rather than HGT.

Answer: we agree with the reviewer’s comment. Despite the difference in GC content is statistically significant, it might be just due to the GC skew. The sentence has been changed as follow: “This could be due to the typical GC skew pattern (Figure 2) or might be indicative of a horizontal gene transfer event” (lines 181-182).

  1. The use of “codon usage”: I have two issues with the discussion of codon usage in the paper. First, at lines 214-217 and in Figure A3, the authors appear to use the term “codon usage” to refer to amino acid frequency in the genome. More typically, “codon usage” refers to the relative proportion of alternative codons for each amino acid. This makes it difficult to interpret Figure A3 and some of the results, because, as far as I could tell, the actual frequency of alternative codons is not considered. I do think the comparison of amino acid usage is still interesting, but this should be clarified.

Answer: we thank the reviewer for the comment. The relative proportion of alternative codons has been calculated and reported in Figure A3 only for the phage, while for the bacteria we report the tRNAs pool. We think that a comparison between the phage codon usage with the tRNA pool from the bacteria is still generally informative. We clarified the difference between the codon usage in the phage and the tRNA pool in the bacteria by modifying the sentence in the Results section as follows “We observed that S144 codon usage correlates with the tRNA pool in S. Muenster and C. sakazakii, with the most required aminoacid (leucine, serine, arginine and valine) corresponding to abundant tRNA in the hosts (Figure A3). “ (lines 243-244).

Second, I do not think the data, as presented, are sufficient to conclude that the potential similarity in amino acid frequency represents a true “adaptation.” To really show this from an evolutionary standpoint, the authors would need to include additional information about the fitness effects of varying the amino acid frequencies. As such, I would suggest restricting this interpretation to the Discussion. Moreover, the authors might consider presenting the data in Figure A3 as a scatterplot with a regression (maybe only choosing one representative host genome for each host species) rather than as a grouped bar plot in order to streamline the presentation and emphasize the strength of the correlation between the genomes.

Answer: As suggested by the reviewer, speculations regarding the adaptation have been removed from the Results section and the Discussion has been modified as follow: “Given the correlation between codon usage and tRNA [60], these data might indicate a certain degree of adaptation of S144 to its hosts. This would be a key adaptation considering that at least one of the two hosts, Salmonella, is translationally biased, i.e. it preferentially uses a subset of codons and their tRNA [61]. Additional experimental data on the fitness effects of varying the amino acid frequencies are necessary to confirm this hypothesis.” In addition, inspired by the Reviewer’s suggestion, we added to Figure A3 a correlation table, summarizing the correlation between phage codon usage and each bacteria tRNAs pool analysed.

  1. The authors should be careful relying on methods like VICTOR for inferring a global phylogeny. Because its approach depends on a whole genome distance metric, it will ignore the potential influence of paralogs and horizontal gene transfer in reconstructing the tree. It might give reasonable potential clades for the purpose of taxonomy, but caution would be needed for deeper inference. It is difficult to assess the accuracy of this kind of tree as compared with a tree based on a collection of core genes (such as could be made using the same genes included in Figure A6).

Answer: we thank the reviewer for the comment and we understand and partially share the concern. Nevertheless, the phylogeny built with VICTOR is in agreement with the current taxonomy and here it has been mainly used to highlight that the ‘Salusvirinae’ phages constitute a separate clade, as confirmed by the phylogeny built using the four genes in Figure 6A.

  1. It would be nice to see additional detail in the Methods on what substitution models were used in CLC for the ML trees in Figure A6.

Answer: trees in Figure A6 were built with the default settings of CLC, using Neighbour joining as construction method and WAG as protein substitution model. These details have been added to the Material and Methods too (lines 698-699).

  1. Figure A1: It might be an issue with the image size within the document, but I have a hard time seeing plaques in the first two panels (the plaques in panel 3 on S394 are visible).

Answer: we agree with the reviewer for his/her suggestion and we took new pictures and changed the figure. In addition, the original pictures have been added to the Supplementary material.

  1. Line 163 and 214 (and elsewhere): This is a very minor point, but in several places the authors mention that the genome “contains no tRNAs,” and I would recommend clarifying to read “tRNA synthesis genes.”

Answer: we thank the reviewer for the suggestion and we changed the text accordingly (lines 176, 243, 354, 467, 518, Figure A3).

  1. Figure A4: The enzyme labels appear to be switched on the left side of panel A, since SspI should have more bands (and also based on the color coding of the dots as described in the caption).

Answer: We concur with the reviewer. We corrected the figure.

  1. Lines 204-206 and again in the Discussion (419-420): the authors discuss promoters that “match” the hosts and are “compatible” and refer to Supplementary material. However, I was unable to determine where in the supplement I was meant to look to find this information. I would also suggest that the authors be more specific in describing what it means to match or be compatible. Are these exact matches?

Answer: the information about the promoters are in the full annotation table that we could upload only as Supplementary material (in the same folder as the original gels, not as Appendix). For promoters, we report the host we selected (Salmonella or Cronobacter) and the score calculated by PhagePromoter. We decided to report all predicted promoters with score higher than 0.5. The following information have been added to the Material and Methods: “Promoters were predicted with PhagePromoter [84], selecting Salmonella or Cronobacter as hosts and manually curated. The promoter host and score calculated by PhagePromoter are available in the full annnotation table (Supplementary material)” (lines 633-635).

  1. Minor writing concerns: Overall, I found the writing clear and easy to follow, but there are a few places where the authors might consider breaking paragraphs into smaller pieces that would be easier for the reader to digest and that would maintain greater focus (e.g. the paragraphs beginning on Line 232 and 461).

Answer: we thank the reviewer for the comment and we changed the suggested paragraphs and others accordingly.
